# A Dirichlet Process Prior Approach for Covariate Selection

**DOI:** 10.3390/e22090948

**Published:** 2020-08-28

**Authors:** Stefano Cabras

**Affiliations:** Department of Statistics, Universidad Carlos III de Madrid, 28903 Madrid, Spain; stefano.cabras@uc3m.es

**Keywords:** covariate inclusion probability, conventional priors, Dirichlet process prior, non-local prior, ordinary linear regression, variable selection

## Abstract

The variable selection problem in general, and specifically for the ordinary linear regression model, is considered in the setup in which the number of covariates is large enough to prevent the exploration of all possible models. In this context, Gibbs-sampling is needed to perform stochastic model exploration to estimate, for instance, the model inclusion probability. We show that under a Bayesian non-parametric prior model for analyzing Gibbs-sampling output, the usual empirical estimator is just the asymptotic version of the expected posterior inclusion probability given the simulation output from Gibbs-sampling. Other posterior conditional estimators of inclusion probabilities can also be considered as related to the latent probabilities distributions on the model space which can be sampled given the observed Gibbs-sampling output. This paper will also compare, in this large model space setup the conventional prior approach against the non-local prior approach used to define the Bayes Factors for model selection. The approach is exposed along with simulation samples and also an application of modeling the Travel and Tourism factors all over the world.

## 1. Introduction

The variable selection problem in regression analysis consists of finding a suitable set of predictors for the response variable *y* from the columns *p* of a fixed design matrix $X_{n\times p}$ with *n* rows (observations). In a Bayesian setting we have to evaluate the posterior probability $\pi (M_{\gamma }|y)$ of each regression model $M_{\gamma }$, where with the usual notation $\gamma =(\gamma_{1},\ldots ,\gamma_{p})$, with $\gamma_{j}=1$ if column $X_{j}$, j=1,…,p is in model $M_{\gamma }$. The aim of this paper is to estimate $\pi (M_{\gamma }|y)$ for all γ∈Γ. The main point is that the cardinality of Γ is of order $2^{p}$, huge even for moderate values of p≈1000 even if it were p<n. In what follows, whatever *p* is, we only evaluate the evidence for models in which the size is no larger than *n* including the intercept and the regression the error variance.

Each $\pi (M_{\gamma }|y)$ can be calculated exactly (not just estimated) by means of all $2^{p}$ Bayes Factors (BF) $BF_{\gamma 0}$, $\pi \left(M_{\gamma }|y\right)=\pi \left(M_{\gamma }\right)BF_{\gamma 0}{\left(1+\sum_{\forall \gamma \neq 0}BF_{\gamma 0}\right)}^{-1}$, where the marginal distribution of model $M_{\gamma }$ is compared against that of a common null nested model $M_{0}$. The reference null model considered here is the regression model with the intercept only. To obtain $\pi (M_{\gamma }|y)$, a prior on model space Γ, $\pi (M_{\gamma })$ must be chosen.

Different priors on Γ have been proposed, starting from the discrete uniform $\pi \left(M_{\gamma }\right)=2^{-p}$ to fulfill the insufficient reasonable principle requirements, up to the hierarchical uniform prior [1]π(Mγ)∝ppγ in which $p_{\gamma }$ indicates the size of model γ. Under the hierarchical uniform, prior models of the same sizes receive the same prior probability and this also controls for the false discovery rate in declaring a covariate important when it is not important. This is relevant for large *p* and when the model is sparse [1].

More than focusing on comparing different definitions of $\pi (M_{\gamma })$, we consider different definitions of $BF_{\gamma 0}$ given by the prior probability of model parameters. In particular, we center on the semi-conjugate multivariate normal on regression coefficients, which leads to closed-form expression of $BF_{\gamma 0}$, which is important for computational feasibility. For instance, the conventional prior approach in [2] has suitable properties (or *desiderata*). An important one is the so-called *predictive matching property*, which assures that $BF_{\gamma 0}=1$ if there is not enough information in the data to distinguish between $M_{\gamma }$ and $M_{0}$. This is obtained by using the concept of effective sample size, which is the number used to rescale the prior covariance matrix to obtain unit information prior, which is a prior that bears the same information as one sample. The other approach is that of the non-local priors detailed in [3]. These priors are non-local in the sense that the minimum prior density is at the null hypothesis in contrast to other usual approaches. The effect is that such a set of priors, asymptotically in *n*, has a larger increment in recognizing the true set of important covariates (the so-called learning rate) more than the conventional prior approach, although they do not meet the predictive matching desiderata. In the sequel, we refer to these two sets of priors as conventional and non-local prior approaches and these are the only two definitions of BFs considered in this work.

Beyond choices of model parameters prior and/or model prior, the point of this work is that even under closed-form expression of BFs an exhaustive exploration of Γ is not feasible and thus a stochastic model search must be employed to obtain an estimation of $\pi (M_{\gamma }|y)$. By estimating $\pi (M_{\gamma }|y)$ and intending to interpret the nature of the true underlying model, a summary of $\pi (M_{\gamma }|y)$ which is of particular interest here is the inclusion probability of a covariate *j*, denoted by τ and defined by marginalizing $\pi (M_{\gamma }|y)$ overall γ such that $\gamma_{j}=1$. τ is necessary to define the *median probability model* [4] defined as the model in which covariates have τ>0.5. If such a model exists, it is proved to be very near to the true model even under strong collinearity [5].

When Γ cannot be fully explored, a stochastic model search is employed in place of heuristic algorithms as they allow to approximate $\pi (M_{\gamma }|y)$. In the case of the normal linear regression model, a popular stochastic approach is that of the well known Gibbs-sampling in [6]. Such an algorithm implements a Markov chain by jumping from one model to another based on the marginal distribution of the data (the same that appears in the Bayes Factors). This algorithm returns a dependent sequence of $M^{\left(S\right)}=\left(\gamma^{\left(1\right)}\in \Gamma ,\ldots ,\gamma^{\left(S\right)}\in \Gamma \right)$ assumed to be a sample of size *S* from $\pi (M_{\gamma }|y)$. The Gibbs-sampling operates on the space Γ using the fact that a closed-form expression of the marginal distributions is available for the normal regression model with the above-mentioned set of conjugate priors (conventional and non-local). The main advantage of Gibbs-sampling over heuristic approaches is that the theory assures that when the number of steps S→∞ all Γ has been properly explored, whereas heuristic methods (like step-wise methods) may stop at local maxima and may not bear properly posterior model uncertainty.

The aim of this paper is to use $M^{\left(S\right)}$ to obtain an estimation of τ, namely $\hat{\tau }$, and in particular the new estimators ${\hat{\tau }}_{b}$ and ${\hat{\tau }}_{f}$, later defined. A different definition of $\hat{\tau }$ derived from the Gibbs-sampling algorithm has been studied in [7] applying the known concept of sampling theory. In particular, two estimators have been fully characterized:(*i*)The intuitive and popular *empirical* proportion of the sampled models in $M^{\left(S\right)}$ containing covariate *j*, ${\hat{\tau }}_{e}=S^{-1}\sum_{\gamma \in M^{\left(S\right)}}^{S}1{\left(\gamma \right)}_{\gamma_{j}=1}$, where $1{\left(\gamma \right)}_{\gamma_{j}=1}$ is the usual indicator function for the scalar $\gamma_{j}=1$ in the vector γ. This estimator is called the empirical estimator;(ii)The *renormalized* proportion of the sampled models containing covariate *j*, ${\hat{\tau }}_{r}=\frac{\sum_{\gamma \in M^{\left(S\right)}}1{\left(\gamma \right)}_{\gamma_{j}=1}\pi \left(M_{\gamma }\right)BF_{\gamma^{\left(s\right)}0}}{\sum_{\gamma \in M^{\left(S\right)}}\pi \left(M_{\gamma }\right)BF_{\gamma^{\left(s\right)}0}}$, called the renormalized estimator.

According to [7] both are consistent estimators of τ for S→∞ and that the error of ${\hat{\tau }}_{r}$ is the sum of two components: the error of ${\hat{\tau }}_{e}$ plus (or minus) a term that depends on the correlation between posterior probability of a $M_{\gamma }$ and its probability to be a visited model for the Gibbs-sampling algorithm. Basically, if the Gibbs-sampling moves very little around a model $M_{\gamma }$ just because it has the highest posterior probability (but not necessarily the largest one), ${\hat{\tau }}_{r}$ will be more biased than ${\hat{\tau }}_{e}$ [7] otherwise it will be more precise than ${\hat{\tau }}_{e}$. Moreover, if the model is sparse, the visited models, for finite *S* depends on the initial state of the chain (for instance: the null model $M_{0}$ or the full model for p<n).

The idea of this paper is to rethink about all these estimators of τ and use a proper Bayesian approach to analyze the sequence $M^{\left(S\right)}$. In practice, $M^{\left(S\right)}$ can be viewed as a sequence of non-ordinal categorical variables with two levels, which gives rise to representing Γ as the set of a *very sparse* contingency table of dimension $2^{p}$ cells, which are the probabilities $\pi (M_{\gamma }|y)$. Therefore, estimating cell probabilities is equivalent to the estimate of the posterior model probabilities. The main observation here is that this is the same setup as in [8] except for the fact that samples $M^{\left(S\right)}$ are dependent instead of being independent as assumed in [8], although for large *S* and given that this is a Gibbs-Sampling, dependence becomes mild. Deriving the posterior distribution of cells probability is the same as deriving an estimation of $\pi (M_{\gamma }|y)$ and thus the τ. That is, from $\pi (\tau |M^{\left(S\right)})$ we define ${\hat{\tau }}_{b}=E_{\pi (\tau |M^{\left(S\right)})}\tau$ and this is one of the proposed estimators that we will compare against ${\hat{\tau }}_{e}$ and ${\hat{\tau }}_{r}$ allowing also to properly account for the uncertainty around the obtained value of $\hat{\tau }$.

The following Section 2 will illustrate the Bayesian model for analyzing $M^{\left(S\right)}$ and its approximation that is used to obtain ${\hat{\tau }}_{b}$ and a ${\hat{\tau }}_{f}$ (specified below). Further, model implementation, simulation study, and real data application on Travel and Tourism data are considered in Section 3. Conclusion and final remarks are left for Section 4.

## 2. The Dirichlet Process Mixture Model for Estimating Posterior Model Probabilities

The Dirichlet process prior model was initially considered for sparse contingency tables in [8] and here it is applied to the more specific context of covariate selection and consequent estimation of covariate inclusion probabilities.

Parameters of interest are $B=\left\{\pi_{\gamma_{1}\gamma_{2}\cdots \gamma_{p}},\gamma_{j}=0,1,j=1,\ldots ,p\right\}\in \Pi$ which is the set of all probabilities tensor on the space Γ of size $2^{p}$. These are the joint probabilities of all covariates and thus of all $M_{\gamma }$ on space Γ, namely the $2^{p}$ cells probabilities, where ${∥B∥}_{1}=\sum_{\gamma_{1}=0}^{2}\cdots \sum_{\gamma_{p}=1}^{d_{p}}\left|\pi_{\gamma_{1}\gamma_{2}\cdots \gamma_{p}}\right|=1$ and every $0\leq \pi_{\gamma_{1}\gamma_{2}\cdots \gamma_{p}}\leq 1$.

In the actual setup, these probabilities are attempted to be estimated with the Gibbs-sampling output, $M^{\left(S\right)}$ regarded here as the data used to estimate B. Specifically, a sample $\gamma_{j}^{\left(s\right)}\in M^{\left(S\right)}$ is a dichotomous unordered categorical variable and we denote the two categories of $\gamma_{j}^{\left(s\right)}$, by $c_{j}=0,1$. The key idea in the Dirichlet process mixture model is representing probability B by decomposing it as an addictive mixture of *k* (possibly infinite) sets of probabilities,
$$B=\sum_{h=1}^{k}v_{h}\Psi_{h},\;\Psi_{h}=\psi_{h}^{\left(1\right)}⊗\psi_{h}^{\left(2\right)}⊗\cdots ⊗\psi_{h}^{\left(p\right)}$$
where $v={\left(v_{1}\geq \ldots \geq v_{k}\right)}^{\prime }$ is the probability vector of the h=1,…,k sets of distributions $\Psi =\left(\Psi_{1}\ldots \Psi_{k}\right)$, with $\Psi_{h}\in \Pi_{1\cdots p}$, where $\psi_{h}^{\left(j\right)}$ is the probability for covariate *j* to be included into the set of predictors given the probability distribution over all Γ labeled by *h*.

It is important to note that $\psi_{h}^{\left(j\right)}$ is the *j* covariate inclusion probability and its estimator is also an estimator of τ if the distribution *h* has high posterior probability $v_{h}$ given $M^{\left(S\right)}$. In particular, if sets of probabilities are ordered according to their posterior probabilities, and $v_{1}\approx 1$ then estimation of $\Psi_{1}^{\left(j\right)}$ could be a good candidate for being a conditional (to h=1) estimator of τ.

For this purpose, we define the estimator
$${\hat{\tau }}_{f}=\Psi_{1},$$
understood for all *j*, given that the first component of the mixture has the highest probability of $v_{1}$.

The likelihood of the two parameters v and Ψ given $M^{\left(S\right)}$ is
$$\mathrm{Pr}\left(\gamma_{1}^{s}=c_{1},\ldots ,\gamma_{p}^{s}=c_{p}|v,\Psi \right)=\pi_{c_{1}\cdots c_{p}}=\sum_{h=1}^{k}v_{h}\prod_{j=1}^{p}\psi_{hc_{j}}^{\left(j\right)}.$$

Introducing the latent class indicator $z_{s}\in \left\{1,\ldots ,k\right\}$, the conditional probability to a specific set *h* is $\mathrm{Pr}\left(\gamma_{j}^{s}=c_{j}|z_{s}=h\right)=\psi_{hc_{j}}^{\left(j\right)}$ and we have that ψs are the inclusion probabilities of covariates conditional to the latent class indicator.

Thus the marginal distribution of τ is obtained by marginalizing over all latent class indicators, namely $\tau =\psi^{\left(j\right)}=\sum_{h=1}^{k}\mathrm{Pr}\left(\gamma_{j}^{s}=1|z_{s}=h\right)\mathrm{Pr}\left(z_{s}=h\right)$, is the parameter of interest which leads to the definition of our estimator
$${\hat{\tau }}_{b}=E_{\pi (\tau |M^{\left(S\right)})}\tau =E\left(\psi^{\left(j\right)}|M^{\left(S\right)}\right)=\sum_{h=1}^{k}\mathrm{Pr}\left(\psi_{hc_{j}}^{\left(j\right)}|M^{\left(S\right)}\right)\mathrm{Pr}\left(v_{h}|M^{\left(S\right)}\right).$$

The larger the *k* is, the better is the representation of B, thus we allow for k=∞ by using the following non-parametric prior:(1)$$\begin{array}{ccc} B & = & \sum_{h=1}^{\infty }v_{h}\Psi_{h},\;\Psi_{h}=\psi_{h}^{\left(1\right)}⊗\cdots ⊗\psi_{h}^{\left(p\right)}\\ \psi_{h}^{\left(j\right)} & ∼ & P_{0j},\;\text{independently}\;\text{for}\;j=1,\ldots ,p\;\mathrm{and}\;h=1,\ldots ,\infty \\ v & ∼ & Q, \end{array}$$
where $P_{0j}$ corresponds to a Dirichlet measure and *Q* to a Dirichlet process.

For the usual stick-breaking stochastic representation of this model, we have
$$\begin{array}{ccc} \gamma_{j}^{s} & ∼ & \mathrm{Multinomial}\left(\left\{0,1\right\},\psi_{z_{i}}^{\left(j\right)},\ldots ,\psi_{z_{i}d_{j}}^{\left(j\right)}\right)\\ z_{i} & ∼ & \sum_{h=1}^{\infty }V_{h}\prod_{l<h}\left(1-V_{l}\right)\delta_{h},\;V_{h}∼beta\left(1,\alpha \right)\\ \psi_{h}^{\left(j\right)} & ∼ & \mathrm{Dirichlet}\left(a_{j1},\ldots ,a_{jc_{j}}\right)\\ \alpha & ∼ & \mathrm{Gamma}\;(a_{\alpha },b_{\alpha }), \end{array}$$
where the Multinomial notation is here over-engineered as $M^{\left(S\right)}$ observations are indicator variables. Parameters $a_{j1}=\ldots =a_{jc_{j}}=1$ induce non informative *a priori* information about the probabilities of each covariate being in the model. α is the usual concentration parameter on the space of latent classes of model probability distributions. That is for small values of α, the probability of having many classes of different probability distributions decreases. In what follows we will consider either α fixed or under another layer of uncertainty by setting a gamma prior with shape $a_{\alpha }$ and scale $b_{\alpha }$ with $a_{\alpha }+b_{\alpha }=1/2$ and $a_{\alpha }=1/4$ as in [8].

With this model at hand, we can obtain justification for the popular empirical estimator of τ. That is, S→∞
${\hat{\tau }}_{b}\approx {\hat{\tau }}_{e}$ as this is a usual regular Bayesian model in which the prior information is washed out by the sample size *S*. This is an asymptotic, although not Bayesian, justification of using ${\hat{\tau }}_{e}$ instead of the proposed ${\hat{\tau }}_{b}$. In this comparison scenario, ${\hat{\tau }}_{f}$ instead plays the role of a more robust estimation as it is based on that distribution which has the largest probability. Therefore, comparing ${\hat{\tau }}_{b}$ against ${\hat{\tau }}_{f}$ is the same as comparing means over modes except that the underlying randomness is on probability distributions instead of random variables. Finally, comparing ${\hat{\tau }}_{b}$ and ${\hat{\tau }}_{f}$ against ${\hat{\tau }}_{e}$ and ${\hat{\tau }}_{r}$ is the same as comparing sampling-based estimators against Bayesian ones which incorporate the shrinkage effect of the prior. The claim here is that such an effect can be arbitrarily large for *S* finite and *p* arbitrary large.

### 2.1. Variational Algorithm for Approximate ${\hat{\tau }}_{b}$

The above stochastic representation suggests to obtain $\pi (M_{\gamma }|y)$ by using another Gibbs-sampling exposed in [8] and it was used to obtain simulations from the posterior distribution of ψs and *v*s. However, this algorithm can be very slow for large *p* and the benefits of the proposed Bayesian approach can be compensated by just calculating ${\hat{\tau }}_{e}$ or ${\hat{\tau }}_{r}$ over larger values of *S* (if these were possible to be obtained).

To avoid this drawback of the proposed modelling approach we make use of a recent and faster variational algorithm illustrated in [9] and also implemented in the R package mixdir. The algorithm relies on using approximated distribution, for the posterior of v and ψ. Such distributions are derived by applying the mean-field theory to variational inference (see [10]). The approximating *q* distributions of the variational approach lay down to be a mixture of Dirichlet distributions. For more details, see [9]. This algorithm is much faster than the initial Gibbs-sampling in [8] and thus may compensate for the need of a larger *S* to obtain good estimations of τ.

## 3. Implementation and Examples

The R implementation of the proposed model is straightforward and it does not even need an ad hoc appendix because major packages are already available. In particular, the implementation requires the following packages for Gibbs-sampling $M^{\left(S\right)}$: BayesVarSel for BF based on the conventional prior approach [2] and/or mombf for BF based on non-local priors [11]. Finally, package mixdir contains the variational Bayes method sketched above in Section 2.1.

The minimal implementation for a response *y* and a design matrix *X* with *p* columns requires two steps:Obtain a sample of $M^{\left(S\right)}$ use:for conventional prior: gammas<- GibbsBvs(y,X)$modelslogBF[,-p+1]for non-local prior: gammas<- modelSelection(y,X)$postSampleEstimate B by cellsprob=mixdir(gammas) and calculate, for generic *j* covariate, the inclusions probabilities in each of the component of the mixture, pp=unlist(cellsprob$category_prob[[j]]). Then${\hat{\tau }}_{b}$ is sum(pp[names(pp)==“1”]*cellsprob$lambda)${\hat{\tau }}_{f}$ is pp[names(pp)==“1”][1]

In the examples and simulation studies illustrated below, we will mainly play with the number of rows *S* of above-calculated matrix gammas.

### 3.1. Riboflavin Simulation Study

In what follows we will consider the Riboflavin dataset (see [12]) related to the riboflavin production by *Bacillus subtilis*. We have n=71 observations and p=4088 predictors (gene expressions) and a one-dimensional response (riboflavin production), *y*. We assume that $BF_{\gamma 0}$ is obtained from the conventional prior and non-local prior. Priors on models, $\pi (M_{\gamma })$ is the Uniform prior.

We use the Riboflavin dataset only for fixing $X_{p}$ and simulating 10,000 times the response vector *y* of size *n* according to $y=\sum_{i=1}^{3}i\times X_{p_{i}}+\epsilon$, ϵ∼N(0,2) and $p_{1},p_{2},p_{3}$ are three columns of $X_{p}$ picked at random for each simulated response (the rest of columns of $X_{p}$ are supposed to have no effect on *y*).

The Gibbs algorithm starts at the null model and *S* has different sizes S=100,500,1000. The goal is to compare the proposed estimator against existing ones.

Results regarding the estimation of τ over all simulations with the Conventional and Non-Local priors are shown in Figure 1.

It is possible to appreciate that when the signal in the data is small (coefficients are 1 or 2) and *S* is not large enough, the proposed estimators ${\hat{\tau }}_{b}$ and ${\hat{\tau }}_{f}$ perform better than the existing ones, while the ${\hat{\tau }}_{r}$ performs better under the non-local prior approach. This is because, as mentioned above, the non-local prior approach favors the model learning rate which is reflected in the values of the BF used to renormalize ${\hat{\tau }}_{e}$.

However, such improvements depend on the specific simulation analysis, that is the design matrix, the noise in the response (here, two standard deviations) the specific value of the assumed coefficients (i.e., 1, 2 and 3) and also the values of *S*. In particular, it seems to disappear when the value of the coefficient is high (i.e., 2 or 3). To generalize these results concerning the choice of specific values of coefficients, regression error standard deviation, and *S*, we analyze the results using probit regressions (one for each prior). In particular, we transform 10,000 estimators $\hat{\tau }$ on the probit scale and regress it with respect to the signal into the data (the value of the coefficient) and estimator type (four in total) with interactions among them (in total we have 12 probit regression coefficients: main effects plus interactions).

The resulting coefficient regressions that we focus on indicate the improvement to the ${\hat{\tau }}_{e}$ and the signal when the coefficient is 1. Such an improvement is an increment in the probit scale of the probability of having $\hat{\tau }=1$ for each combination of coefficient and estimator type. These increments are between 0.62 and 2.14 (all highly significant) and are important if the original point in the probit scale is low, while they are negligible if the point on the probit scale is large. Such an initial point corresponds to a signal in the data and the value of *S*. The increment on $\hat{\tau }$ and the signal in the data is reported in Figure 2.

From Figure 2 we can see that the proposed estimators ${\hat{\tau }}_{b}$ and ${\hat{\tau }}_{f}$ generally outperform the existing estimator and popular ${\hat{\tau }}_{e}$ as the increment on $\hat{\tau }$ is larger when the signal in the data is lower. The ${\hat{\tau }}_{r}$ perform better than ${\hat{\tau }}_{f}$ under the non-local prior as the corresponding BFs are very informative for detecting variables.

### 3.2. Travel and Tourism Data Set

In this application, we want to explore the determinants of Travel and Tourism (T&T) global expenditure using data of the World Economic Forum, based on the latest Travel and Tourism Competitiveness Report (2019). The statistical unit is the country and the response variable is the log of total tourism expenditure, obtained by multiplying the number of arrivals by the reported individual expenditure. Data are obtained at http://www3.weforum.org/docs/WEF_TTCR19_data_for_download.xlsx and further filtered to have a complete dataset https://raw.githubusercontent.com/scabras/varseldmmp/master/tourism-data.csv The actual code to analyze it is here: https://github.com/scabras/varseldmmp/blob/master/s-example-tourism.md.

Tourism worldwide is threatened by the Covid-19 spread and it represents an important source of income for many countries. For instance, in Spain, Mexico, and France, according to The Organization for Economic Co-operation and Development (OECD) in 2016, tourism represented more than 7% of the Gross Domestic Product. The total number of covariates considered here is p=65 fully observed in n=52 countries.

We apply the analysis using Conventional and non-local priors assuming that the number of latent probability distributions on the model space is k=10 and a fixed concentration parameter α=1 to be as parsimonious as possible on determining the median probability model for S=1000 and a burn-in of 100 Gibbs-sampling steps.

It is important to note that to reach almost a substantial agreement between the actual estimators ${\hat{\tau }}_{e}$, ${\hat{\tau }}_{r}$ on one side and the newly proposed ones ${\hat{\tau }}_{b}$, ${\hat{\tau }}_{f}$ it is necessary to have *S* = 10,000, which corresponds to about seven times more the computational time used to obtain Table 1 which reports the estimated median probability models.

The most important factor is precisely the one most affected by Covid-19, which is the number of operating airlines. The other factors are less related to Covid-19 but are still important. One is the organizations of the T&T govern monitoring represented by the timeliness of providing data on T&T. There are also variables related to country features: infrastructures (connection and power) internet search of T&T resources and the presence of natural resources (total known species). As observed above, non-local priors lead to more complex models than the conventional prior, but looking at the proposed estimators they seem to be more robust with respect to the choice of the prior on model parameters, in fact, ${\hat{\tau }}_{b}$ reports almost the same covariates regardless of the BF definition.

## 4. Remarks

This work aims to analyze the Gibbs-sampling output of model exploration according to a genuine Bayesian analysis and avoid frequentist approaches that are of a critical application for *p* large and *S* finite. Further, this is also the first work on comparing the conventional versus the non-local prior definition of BFs, which represent at the moment the state of the art in covariate selection under the normal linear model. The posterior mean of the τ is estimated with ${\hat{\tau }}_{b}$ results to be a robust estimator of the τ. This is because of the Dirichlet model, which shrinks to zero some of the observed empirical proportions, resulting from ${\hat{\tau }}_{e}$, and increases the others. Moreover, it is also clear that the proposed ${\hat{\tau }}_{r}$, studied only for conventional prior in [7], seems to work very well under the improved learning rate of the non-local prior approach.

As a further line of investigation, but beyond the scope of this paper, we note that $\pi (M_{\gamma })$ (here, the uniform) can be substituted by the Dirichlet process model illustrated above and thus used directly as $\pi (M_{\gamma })$ by incorporating it into the Gibbs-sampling. This is important in order to match B in (Equation 1) with $\pi (M_{\gamma })$. This is incidentally done in [7], with the uniform $\pi (M_{\gamma })$ when using ${\hat{\tau }}_{r}$. In general, the problem of model exploration would deserve more Bayesian methods than the existing ones. 

## Figures and Tables

**Figure 1 entropy-22-00948-f001:**
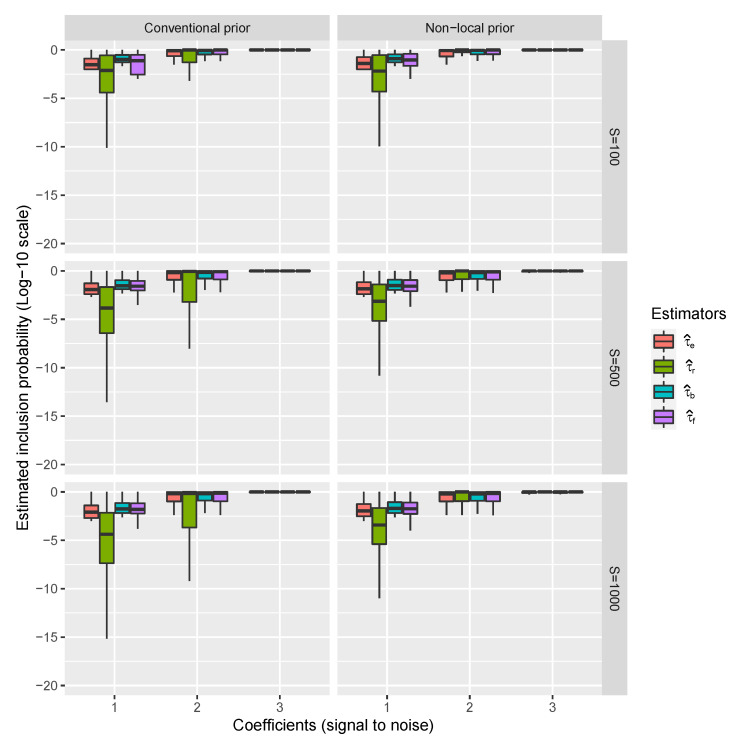
Riboflavin simulation results. Distributions of ${\hat{\tau }}_{e}$, ${\hat{\tau }}_{r}$, ${\hat{\tau }}_{b}$ and ${\hat{\tau }}_{f}$ over the 10,000 datasets for only covariates inside the model, ordered by the magnitude of their corresponding coefficients and the prior used to calculate BFs. Coefficients also represent the signal into the data (i.e., the value of the coefficients) with respect to the residual standard error (the noise).

**Figure 2 entropy-22-00948-f002:**
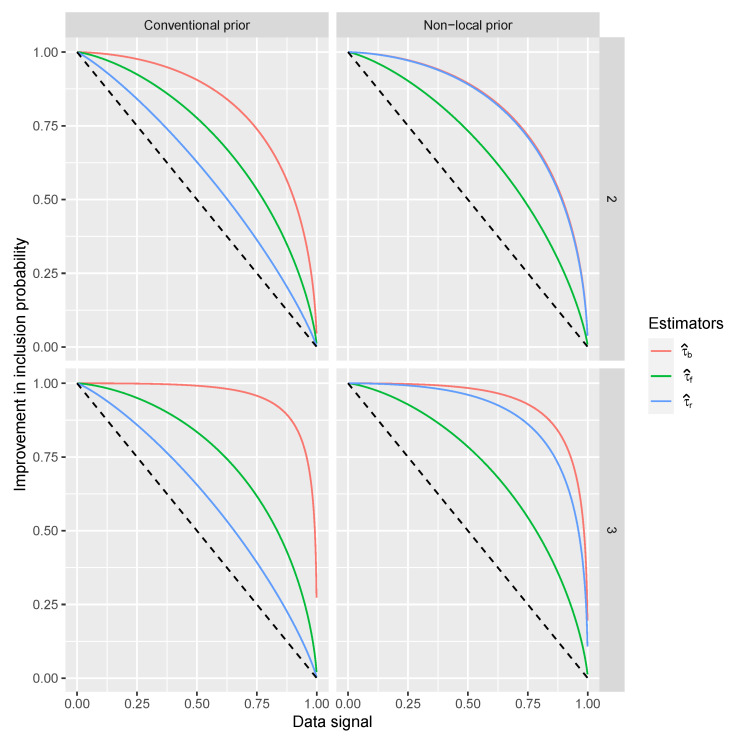
Riboflavin simulation results. Estimate improvements into the inclusion probabilities (vertical axis) with respect to the signal in the data and *S* (horizontal axis) for ${\hat{\tau }}_{e}$ (dashed line), ${\hat{\tau }}_{r}$, ${\hat{\tau }}_{b}$ and ${\hat{\tau }}_{f}$ estimated over the 10,000 datasets. For only covariates inside the model, ordered by the magnitude of their corresponding coefficients and the prior used to calculate BFs.

**Table 1 entropy-22-00948-t001:** Tourism and Travel (T&T) data set. Estimated median probability models according to the different estimators $\hat{\tau }$ crossed with BF definitions given by conventional and non-local priors. In parenthesis, $\hat{\tau }$ is reported when the covariate is in the median probability model.

*Name of Covariate*	Conventional Prior	Non-Local Prior
τ^e	τ^r	τ^b	τ^f	τ^e	τ^r	τ^b	τ^f
number of operating airlines	(1.00)	(1.00)	(1.00)	(1.00)	(1.00)	(1.00)	(1.00)	(1.00)
timeliness of providing								
monthly/quarterly t&t data	(0.95)	(1.00)	(0.74)		(0.93)	(1.00)	(0.83)	
individuals using internet, %		(0.91)	(0.57)		(0.76)		(0.72)	
purchasing power parity		(0.96)	(0.51)		(0.64)		(0.53)	
natural tourism digital demand		(0.75)			(0.68)		(0.61)	
active mobile broadband internet								
subscriptions/100 population		(0.82)						
openness of bilateral air service								
agreements		(0.96)						
total known species		(0.71)

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
