# Peer review of "A Dirichlet Process Prior Approach for Covariate Selection"

_entropy, 2020, doi:10.3390/e22090948_

Round 1

Reviewer 1 Report

This paper proposes a new Bayesian approach for model selection in linear regression. The authors employ the Dirichlet process mixture model for estimating posterior model probabilities. They provided two types of estimators of model probabilities and demonstrated their performance through a simulation study and data analysis. I think this paper is worth publishing, but there are some issues to be addressed before acceptance. 

  1. I found a lot of typos in the manuscript. For example, "evaluated" (page 1, line 19) and "wherewith" (page 1, line 20). I recommend the authors carefully check all the sentences before resubmission. 
  2. (page 2, line 44-45) It would be very helpful for potential readers to explain little more details of "non-local priors".
  3. (page 2, line 69) Please define "1(\gamma)".
  4. (page 3, line 107) There is some unnecessary space before "where". The same happens on page 4, line 123.
  5. (page 3, line 113) I suppose \Psi_1 rather than \psi_1 in the definition of \hat{tau}_f.
  6. (page 6, figure 1) What do you mean by "signal to noise"? I think each value of the x-axis corresponds to the true values of the regression coefficients. Regarding the y-axis, I suppose the use of log-scale can be more interpretable than showing log-transformed values.   
  7. (Section 3.2) It would be interesting to compare the computation time of the proposed method and that of existing methods. Since the authors are concerned with computation time in Section 2.1, such information would be helpful. 
  8. (Reference) Some details of the 3rd reference (e.g. journal name) seem missing.

Author Response

I thank the referee for the useful comments which made more clear the exposition of the ideas. Here is a point-to-point answer:

  1. Ref: I found a lot of typos in the manuscript. For example, "evaluated" (page 1, line 19) and "wherewith" (page 1, line 20). I recommend the authors carefully check all the sentences before resubmission. 

Ans: These and other typos have been checked.

2. Ref: (page 2, line 44-45) It would be very helpful for potential readers to explain little more details of "non-local priors".

Ans: The following sentence has been added: "These priors are non-local in the sense that put minimum prior density at the null hypotheses in contrast to other usual approaches. The effect is that such a set of priors..."

Ref: (page 2, line 69) Please define "1(\gamma)".

Ans: We added the definition: " where $\I(\gamma)_{\gamma_j=1}$ is the usual indicator function for the scalar $\gamma_j=1$ in the vector $\gamma$.".

Ref: (page 3, line 107) There is some unnecessary space before "where". The same happens on page 4, line 123.

Ans: These spaces are due to the latex formatting macro used by the journal.

Ref: (page 3, line 113) I suppose \Psi_1 rather than \psi_1 in the definition of \hat{tau}_f.

Ans: \psi_1 has been changed in \Psi_1 

Ref: (page 6, figure 1) What do you mean by "signal to noise"? I think each value of the x-axis corresponds to the true values of the regression coefficients. Regarding the y-axis, I suppose the use of log-scale can be more interpretable than showing log-transformed values.

Ans: In the caption of the Figure we added the following sentence "Coefficients also represent the signal into the data (i.e. the value of the coefficients) with respect to the residual standard error (the noise ).". The y-axis reports the estimated value on the log-10 scale.

Ref: (Section 3.2) It would be interesting to compare the computation time of the proposed method and that of existing methods. Since the authors are concerned with computation time in Section 2.1, such information would be helpful.

Ans: Thank you for this important point. We conducted a study in which we highlight that to reach almost a substantial agreement between the actual estimators and the newly proposed ones it is necessary to have $S=10000$ which corresponds to about seven times more the computational time actually used to obtain Table 1.

Ref: (Reference) Some details of the 3rd reference (e.g. journal name) seem missing.

Ans: This is a CRAN package and now it is highlighted.

Reviewer 2 Report

This manuscript tests different selections of the model priors and Bayes factors and aims to estimate inclusion probability. This study can be important in selecting an important subset of predictors when their size is large. The author used the MCMC stochastic approaches to estimate the probability of selecting a model given the data. Overall, this manuscript is well written with two examples. This reviewer provides minor comments.

1. Page 1, Line 28. Typo

2. Page 2. Lines 59-60. The author may provide some reasons and rationale behind this idea. For example, what is the advantage of using the Gibbs sampling techniques for linear regression problems?

3. Page 5. Lines 147 and 163. Typos

4. Page 5. It is not clear how the number of parameter p in the original datasets is modified to 10,000.

Author Response

I thank the referee for the comments. Here is a point-to-point answer to the suggested correction:

Ref: Page 1, Line 28. Typo

Ans: The typo has been corrected. The sentence is "The here considered reference null model is the regression model with the intercept only"

Ref: Page 2. Lines 59-60. The author may provide some reasons and rationale behind this idea. For example, what is the advantage of using the Gibbs sampling techniques for linear regression problems?

Ans: The following paragraph has been added "The Gibbs-sampling operates on the space of $\gamma$ using the fact that closed-form expression of the marginal distributions is available for the normal regression model with the above-mentioned set of conjugate priors (conventional and non-local). The main advantage of Gibbs-sampling over heuristic approaches is that theory assures that when the number of steps $S \rightarrow \infty$ all $\Gamma$ has been properly explored, were as heuristic methods (like step-wise methods) may stop at local maxima and may not bear properly posterior model uncertainty."

Ref: Page 5. Lines 147 and 163. Typos

Ans: L147: changed in ". Such distributions are derived by applying the mean-field theory", L163: changed in "calculate, for generic $j$ covariate, the inclusions probabilities".

Ref: Page 5. It is not clear how the number of parameters p in the original datasets is modified to 10,000.

Ans: All the paragraph has been rephrased and now it appears as "We use the Riboflavin dataset only for fixing $X_p$ and simulating 10000 times the response vector $y$ of size $n$ according to $y=\sum_{i=1}^{3}{i\times X_{p_i}}+\epsilon$, $\epsilon\sim N(0,2)$ and $p_1,p_2,p_3$ are 3 columns of $X_p$ picked at random for each simulated response (the rest of columns of $X_p$ are supposed to have no effect on $y$).".